# Visceral Artery Aneurysms Embolization and Other Interventional Options: State of the Art and New Perspectives

**DOI:** 10.3390/jcm10112520

**Published:** 2021-06-07

**Authors:** Massimo Venturini, Filippo Piacentino, Andrea Coppola, Valeria Bettoni, Edoardo Macchi, Giuseppe De Marchi, Marco Curti, Christian Ossola, Paolo Marra, Anna Palmisano, Alberta Cappelli, Antonio Basile, Rita Golfieri, Francesco De Cobelli, Gabriele Piffaretti, Matteo Tozzi, Giulio Carcano, Federico Fontana

**Affiliations:** 1Diagnostic and Interventional Radiology Department, Circolo Hospital, ASST Sette Laghi, 21100 Varese, Italy; filippo.piacentino@asst-settelaghi.it (F.P.); andrea.coppola@asst-settelaghi.it (A.C.); valeria.bettoni@asst-settelaghi.it (V.B.); edoardo.macchi@asst-settelaghi.it (E.M.); giuseppe.demarchi@asst-settelaghi.it (G.D.M.); federico.fontana@uninsubria.it (F.F.); 2Department of Medicine and Surgery, Insubria University, 21100 Varese, Italy; curti.marco.33@gmail.com (M.C.); c.ossola7@gmail.com (C.O.); gabriele.piffaretti@uninsubria.it (G.P.); matteo.tozzi@uninsubria.it (M.T.); giulio.carcano@uninsubria.it (G.C.); 3Department of Diagnostic Radiology, Giovanni XXIII Hospital, Milano-Bicocca University, 24127 Bergamo, Italy; pmarra@asst-pg23.it; 4Department of Radiology, IRCCS San Raffaele Scientific Institute, San Raffaele School of Medicine Vita-Salute University, 20132 Milan, Italy; anna.palmisano@hsr.it (A.P.); francesco.decobelli@hsr.it (F.D.C.); 5Department of Radiology, IRCCS Azienda Ospedaliero-Universitaria di Bologna, 40138 Bologna, Italy; alberta.cappelli@aosp.bo.it (A.C.); rita.golfieri@unibo.it (R.G.); 6Department of Medical and Surgical Sciences and Advanced Technologies, Radiodiagnostic and Radiotherapy Unit, University Hospital “Policlinico-Vittorio Emanuele”, 95123 Catania, Italy; basile.antonello73@gmail.com; 7Vascular Surgery Department, Circolo Hospital, ASST Sette Laghi, 21100 Varese, Italy; 8Department of General, Emergency and Transplants Surgery, Circolo Hospital, ASST Sette Laghi, 21100 Varese, Italy

**Keywords:** visceral aneurysm, endovascular treatment, embolization, coiling, covered stent, flow-diverting stent

## Abstract

Visceral artery aneurysms (VAAs) are rare, usually asymptomatic and incidentally discovered during a routine radiological examination. Shared guidelines suggest their treatment in the following conditions: VAAs with diameter larger than 2 cm, or 3 times exceeding the target artery; VAAs with a progressive growth of at least 0.5 cm per year; symptomatic or ruptured VAAs. Endovascular treatment, less burdened by morbidity and mortality than surgery, is generally the preferred option. Selection of the best strategy depends on the visceral artery involved, aneurysm characteristics, the clinical scenario and the operator’s experience. Tortuosity of VAAs almost always makes embolization the only technically feasible option. The present narrative review reports state of the art and new perspectives on the main endovascular and other interventional options in the treatment of VAAs. Embolization techniques and materials, use of covered and flow-diverting stents and percutaneous approaches are accurately analyzed based on the current literature. Visceral artery-related considerations and targeted approaches are also provided and discussed.

## 1. Introduction

Visceral artery aneurysms (VAAs), or true aneurysms, are an uncommon entity, with a variable incidence at autopsy finding from 0.01 to 0.2% [1,2]. Several diseases are implicated in vascular damage with possible formation of VAAs, such as atherosclerosis, hypertension [3], medial degeneration [4], arteritis [5,6], infection [7], fibromuscular dysplasia [8], diabetes [9], and congenital anomalies such Ehlers-Danlos syndrome [10]. VAAs usually affect the celiac trunk, the superior mesenteric and inferior mesenteric arteries and their branches, with the splenic artery involved in up to 60% of cases, followed by the hepatic artery [11]. VAAs are usually asymptomatic and incidentally discovered during routine abdominal imaging [11]. The majority of VAAs remain stable over time, but a minority of them that grow present a high risk of rupture [12]. Conditions associated with a higher risk of rupture are pregnancy due to hormonal changes and increased blood flow [13], and the presence of visceral artery pseudoaneurysms or false aneurysms that can be caused by pancreatitis [14], trauma [15], previous surgical or interventional procedures [16,17,18]. Differently from true aneurysms, pseudoaneurysms do not contain all three layers of the vascular wall, and are therefore more fragile. The choice between conservative versus interventional management of VAAs is influenced by different factors, such as patient age, aneurysm size and the involved visceral artery and associated comorbidities. Consensus guidelines suggest treatment in the following situations: asymptomatic VAAs with a diameter larger than 2 cm, or 3 times exceeding the target artery’s size; VAAs with a progressive growth of at least 0.5 cm per year; symptomatic or ruptured VAAs [19,20,21]. Treatment options include surgical, endovascular or percutaneous approaches. Surgery performed open, laparoscopically or robotically, relies on different strategies, including vessel ligation, resection, end to end anastomosis, re-implantation or graft interposition [22,23,24,25]. Endovascular treatment, being less burdened by morbidity and mortality than surgery, is usually considered the first therapeutic option [26,27,28,29], offering different interventional options. Transcatheter embolization with different materials and devices is largely used worldwide. In particular, coil embolization, stent- or balloon-assisted in selected cases, represent the most widespread embolization techniques. Endovascular repair using a covered stent allows for the exclusion of the aneurysm, preserving the flow through the affected visceral artery, thus reducing the potential risk of target organ ischemia; however, its use is challenged by the tortuosity of the visceral arteries. Flow-diverter stents that have been recently introduced for neuro-interventional procedures, represent a promising albeit expensive solution to maintain patency of both the target artery and side branches arising from the VAA. Superficial VAAs can be treated percutaneously with injection of thrombin or other embolic agents under imaging (US, CT) guidance. The aim of the present review is to analyze state of the art and new perspectives of the different embolization techniques, including the other endovascular and interventional options for the treatment of visceral and renal artery aneurysms.

## 2. Imaging: Diagnosis and Post Treatment Monitoring

Several imaging modalities can be used to diagnose VAAs, mostly performed cases in asymptomatic patients. Indeed, the detection of VAAs is frequently circumstantial during routine radiological exams [11]. X-rays are rarely helpful, but may accidentally show calcifications in atherosclerotic VAAs, for example in splenic artery aneurysms [30]. Color Doppler Ultrasonography (CDU), a simple and non-invasive technique, is often used as initial screening tool for abdominal aortic aneurysms, but its diagnostic accuracy is limited in VAAs: bowel gas and fat hinder visceral artery visualization. Low-frequency (2–5 MHz) convex probes may show VAAs as ovoid or round anechoic structures which fill with color, often with a bidirectional waveform pattern [31]. Contrast-enhanced multidetector computed tomography angiography (CTA) is the most commonly used and sensitive technique to diagnose VAAs. Multiphase scans (pre-contrast, arterial phase, venous phase, delayed phase) allow for the panoramic assessment and precise characterization of VAAs. The arterial phase is usually obtained 20–30 s after the intravenous injection of 80–120 iodinated contrast material at a rate of 3–5 mL/s [32]. The arterial phase is crucial not only for the anatomical definition of the vascular lesion, but also to prove and locate contrast extravasation. The rapid acquisition time and the high spatial resolution also makes CTA the gold standard technique in emergency cases of ruptured VAAs, showing extravasation of contrast, in emergency settings in symptomatic patients. Moreover, CTA provides useful information for the planning of the endovascular treatment (embolization, stent-graft, other interventional options) thanks to multiplanar and 3D image reconstruction [33]. CTA-derived information can influence the therapeutic choice and the procedural strategy: the visceral artery involved, vessel tortuosity, aneurysm diameter, aneurysm morphology (saccular vs. fusiform), aneurysm characteristics (calcifications, thrombosis, dissection presence vs. absence), neck (wide vs. narrow), number of efferent/afferent vessels, angle of origin of the target artery from the abdominal aorta (Figure 1). 

Magnetic Resonance Imaging (MRI) allows for the characterization of vascular flow, even without the administration of contrast agents (flow sensitive, time of flight, angiographic sequences) [34]. Furthermore, MRI does not use ionizing radiations and employs less nephrotoxic contrast agent compared with CTA. Up-to-date MRI systems allow for the acquisition of rapid breath-hold vascular imaging sequences and flow quantification with phase-contrast techniques [35]. When incorporated to the 3D Magnetic Resonance Angiography (MRA), 3D Magnetic Resonance Angiography (MRA) has shown huge potential in the characterization of VAAs thanks to high contrast resolution: image contrast is based on a marked reduction of T1 relaxation time in the vascular system as compared to other tissues, and is caused by the injection of a small amount of contrast agent. Disadvantages of MRI (and MRA) are still long acquisition times, which makes it unsuitable in the emergency setting, and the need for patient compliance. Claustrophobia and pacemakers are significant contraindications. Time-Resolved MRA has been reported as a valuable option for non-invasive follow-up of VAAs after coil embolization. The main advantages of MRA compared to CTA are less susceptibility to metallic artifacts from platinum coils and the ability to detect blood flow with a high temporal resolution [36,37]. Metallic and bowel artifacts also make post embolization assessment difficult with CDUS or contrast-enhanced ultrasonography (CEUS). CEUS can be useful for detect an ischemic complication of the target organ, such as the spleen, kidney or liver [38]. 

## 3. Embolization: Techniques and Materials

### 3.1. Coil Embolization

Coil embolization represents the most used endovascular technique to treat VAAs. 

Metallic coils can be used alone or with other embolic agents/devices. Coils can determine mechanical obstruction and secondary thrombosis both by their thrombogenic fibers and induced inflammatory reaction [39]. Fibered coils are considered more thrombogenic than non-fibered coils [40]. Coils should be used in the appropriate size, usually exceeding the vessel lumen by 20%: undersized coils increase the risk of inadequate occlusion or distal migration [41], while oversized coils cannot assume their conformation, minimizing their thrombogenic action [42]. The embolization of middle or large-sized vessels can present pitfalls due to a high risk of coil migration: a particular design characterizes penumbra occlusion device (POD; Penumbra Inc., Alameda, CA, USA) coil systems that have the ability to anchor themselves in large-caliber arteries, preventing the risk of distal migration [43]. High-density coil packing is essential to obtain a permanent aneurysm exclusion. Different coils can be used to embolize VAAs: 0.035-inch coils using 4-French standard angiographic catheters, 0.018-inch microcoils using coaxial microcatheters, and, recently borrowed from interventional neuroradiology, 0.010-inch microcoils in the case of very small branches of visceral arteries [44]. Microcatheters of different diameter and profile associated with microcoils are preferred to standard angiographic catheters due to their better trackability through the tortuous visceral arteries. In the past, pushable coils were largely used due to their much more sustainable cost compared to detachable coils. Nowadays, the cost of detachable coils has dropped significantly, and they present several advantages such as the possibility to be recaptured in case of wrong placement and availability of longer lengths (>50 cm), resulting in coil number sparing. The strategy of coil embolization depends on VAA morphology, size, position (distal vs. proximal) and on the involved artery. VAA exclusion can be obtained via total occlusion or with preservation of the target artery. In the case of fusiform VAAs, which cover the entire circumference of the vessel, a sacrifice of the target artery is usually necessary, while in case of saccular VAAs which involve only one side of the vessel, the preservation of the target artery is usually possible [39,45], provided that the VAA neck is sufficiently narrow. The angiographic procedure is generally performed via a transfemoral approach and involves the catheterization of the main trunk of the target artery with a standard angiographic catheter (4–5 French). After a preliminary diagnostic angiography, a coaxial microcatheter (<3 French) or the standard angiographic catheter is advanced over a guidewire toward the VAA. In case of fusiform VAAs, according to the “isolation technique” [46], the efferent vessel (or vessels), often the aneurysm sac and the afferent vessel are subsequently coil-packed during progressive retraction of the microcatheter to simultaneously exclude the aneurysm and occlude the target artery (Figure 2).

In case of saccular VAAs with a narrow neck, according to the “sac packing technique” [47], the aneurysm sac is progressively filled with coils up to neck to exclude the aneurysm and maintain the patency of the target artery (Figure 3).

Therefore, the risk of distal ischemia is possible only in case of parent vessel sacrifice. This risk is higher in fusiform renal aneurysms, lower in splenic aneurysms due to the ease of collateral developing [48]. Clinically significant complications after VAAs embolization have been rarely reported. End-organ infarcts that occurred, have, in most cases, been conservatively treated [49]. 

### 3.2. Stent-Assisted Coil Embolization

Saccular aneurysms can be further classified into narrow or wide neck according to the entry portal from the parent artery to the aneurysm sac [50]. Uncovered stent-assisted coil embolization strategy is usually employed in wide neck saccular aneurysms that involve a visceral artery with high risk of organ ischemia in case of total vessel sacrifice (Figure 4). 

This technique is employed to prevent coil protrusion and migration from the aneurysm neck, as may occur in renal artery aneurysms [50,51,52]. A long-armed introducer or a guiding-catheter (6-French) is placed at the origin of the renal artery and, after the advancement of the guidewire beyond the aneurysm, the stent is usually implanted before coil placement. In selected cases with thin target arteries, coils may be placed before stent release. In complex anatomic situations with aneurysms in proximity to vessel bifurcations, two uncovered stents may be also implanted in a Y-configuration using a double approach. Subsequently, a microcatheter is placed into the aneurysm sac through the stent mesh and the aneurysm is filled with coils. As alternative aneurysm catheterization may be performed before stent release and in case of complex anatomy (Y-configuration) two uncovered stents may be also implanted using a double approach. Self-expandable stents are more suitable considering the tortuosity of VAAs. A double antiplatelet therapy is recommended to prevent stent-thrombosis as in the case of all VAAs treated with covered stent. Neuro-interventional devices such as retrievable stents according to the waffle cone technique [53], have been used for embolization of renal wide neck aneurysms with the stent-assisted coil embolization strategy [54]. The advantage of this modality is the possibility to avoid definitive stent implantation, that would require long-term antiplatelet therapy.

### 3.3. Balloon-Assisted Coil Embolization

Balloon-assisted coil embolization strategy is also employed in case of wide neck saccular aneurysms [55]. In this situation, a microcatheter is placed within the aneurysm sac while a parallel balloon catheter is inflated along the parent vessel across the aneurysm neck to avoid coils prolapsing into the target artery lumen [56]. Subsequent coil deployment and balloon inflation/deflation allows achieving high-density coil packing minimizing the risk of coil protrusion/migration (Figure 5). 

Compliant balloon catheters used in the balloon-assisted coil embolization of intracranial aneurysms, including different devices supported by 0.010–0.014 guidewires platforms, are particularly suitable for VAAs. A double approach may be necessary for balloon-assisted coil embolization, but a single approach is also possible using recent devices for neurointerventions such as double-lumen balloon catheters [57]. 

### 3.4. Embolization with Adhesive Liquid Embolic Agents (Glue)

Embolization with adhesive liquid embolic agents is used in selected cases, especially in pseudoaneurysms [58,59]. Its immediate action makes it indicated in case of active bleeding and emerging procedures. N-Butyl Cyanoacrylate (NBCA), simply known as glue, is mixed with iodized oil (lipiodol) in variable ratios (1:1 up to 1:8) to control glue polymerization time and to impart radiopacity to the mixture. Before the injection of the NBCA mixture, the death space of the microcatheter is flushed with 5–10% dextrose to prevent polymerization in the device lumen. The NBCA mixture is preferably injected through a 1 or 2.5-mLsyringe and under careful fluoroscopic monitoring. NBCA provides rapid polymerization when in contact with blood and permanent embolization. Immediately after injection, the microcatheter is quickly removed to prevent catheter tip entrapment in the vessel. NBCA is usually less used than coils, especially in VAAs embolization, due to handling difficulty, high risk of non-target embolization with potential complications related to ischemic injury. Non-target embolization may occur during microcatheter retraction due to glue residue stuck to the microcatheter tip [60].

### 3.5. Embolization with Non-Adhesive Liquid Embolic Agents (Onyx, Squid, Phil)

In the past, non-adhesive liquid embolic agents have been widely used in intracranial district for cerebral aneurysms [61], but rarely in abdominal district for VAAs [62]. Nowadays, ethylene vinyl alcohol copolymer (EVOH) agents, well-known as Onyx (Medtronic, Dublin, Ireland), Squid (Emboflu, Gland, Switzerland) and Phil (Precipitating Hydrophobic Injectable Liquid, MicroVention, Terumo, Austin, USA), are routinely used in many abdominal diseases [63,64,65,66,67] and also in splenic [68], hepatic [69], mesenteric [70] and renal aneurysms [71]. These liquid embolic agents are EVOH copolymers mixed with micronized tantalum powder and dissolved in dimethyl sulfoxide (DMSO). EVOH liquid embolic agents have a characteristic “lava-like” consistency [72] with significant advantages over glue, including a viscous nature, slow polymerization time, high fluoroscopic visibility, capability to homogeneously fill the vessel, non-adherence to the microcatheter tip with consequent lower risk of non-target embolization. Compared to glue, disadvantages of EVOH embolic agents are the cost, the need of DMSO-compatible microcatheters, the longer preparation time that requires 20 min of shaking and the DMSO infusion required before injection. DMSO can determine vasospasm and pain due to its endothelial toxicity. EVOH embolic agents are usually injected through a 1 or 2.5-mL syringe gradually pulling back the microcatheter to avoid tip entrapment. Detachable tip microcatheters can be used in particular anatomic situations, although they are more suitable for the use with adhesive glues. EVOH embolic agents can be used combined with other embolic devices to enhance their occlusion capacity (Figure 6) or alone (Figure 7). 

Venturini et al. have recently reported the use of Squid and detachable coils to embolize splenic and renal artery aneurysms: they achieved a totally occlusive cast, minimizing the risk of aneurysm revascularization [68,71]. Squid presents some advantages over Onyx: a less tantalum percentage associated with smaller particles reduces artifacts at CT follow-up. Moreover, Squid is also available in the less viscous formulation 12, which allows a more distal spread in small vessels, particularly useful in arteriovenous malformations [73,74]. Phil is the most recent of the EVOH embolic agents and still lacks a solid grounding in literature, but seems to present some advantages over Onyx (and Squid): fewer CT-artifacts due to the use of iodine contrast instead of tantalum, quicker preparation without shaking, “toothpaste-like” instead of “lava-like” consistency which allows a quicker injection, yellow color instead of black powdered material which does not stain the skin in case of embolization of superficial lesions [67,75]. The embolizing power of Onyx and Squid is likely higher than Phil, although comparative studies are lacking. 

### 3.6. Embolization with Plugs, Microplugs

The Amplatzer Vascular Plug (Abbott, St. Paul, MN, USA) is a conventional embolic device disk made up of a braided nitinol mesh. The plug can be retrieved and readjusted as needed before its final release. The obstacle to blood flow caused by nitinol mesh promotes clot formation. Its thrombogenicity is increased by multiple braid layers [76]. Plugs are widely used in endovascular therapy [77]. The Amplatzer Vascular Plug family is based on four models that fit to the different anatomic and hemodynamic situations: plug IV, used in 4-French catheters, is usually oversized by 20 to 30% of the estimated diameter of the vessel. Plugs IV may be used in VAAs embolization, often combined with other embolic agents, to occlude the main afferent artery for example of giant VAAs [78,79]. Micro vascular plugs (Medtronic, Minneapolis, MN, USA) have the significant advantage that can be placed using 0.021-inch or 0.027-inch microcatheters, but the limit to be suitable for relatively small arteries up to 7–7.5 mm [80]. 

### 3.7. Embolization with Particles

Particles are rarely used for embolization of VAAs due to the risk of distal ischemia. However, in particular scenarios (emergency procedures, ruptured pseudoaneurysms, aneurysms involving terminal branches) polivinyl alcohol particles or embospheres may be employed [81]. Re-absorbable embolic agents such as gelfoam are rarely used, sometimes in association with other embolic agents [82]. 

## 4. Covered Stent

In contrast to embolization, which may present a potential risk of distal ischemia, the endovascular repair with covered stent can exclude the VAA preserving the flow through the affected visceral artery. However, if the endovascular treatment of VAAs is almost always possible by embolization, treatment with covered stent is feasible only in selected cases, when VAAs affect proximal or middle segments of the target artery: in a large reported series of 100 patients affected by VAAs (or visceral pseudoaneurysms) submitted to endovascular treatment, covered stent was feasible in 30% of the cases, while embolization was performed in the remaining 70% of the cases [48]. The elevated tortuosity and the reduced caliber of the visceral arteries usually represent the main limitation to covered stent placement: stable carrier systems (7–8 French guiding catheters or long-armed introducers) at the origin of the target artery, hydrophilic guidewires to engage the efferent vessels, stiff guidewires to make easier the covered stent advancement can facilitate the procedure. Covered stent can be also used in an emergency by experienced operators especially in cases of high risk of organ ischemia as seen in ruptured aneurysms of the hepatic artery (Figure 8). 

Balloon-expandable and self-expandable covered stents are widely used in endovascular repair of VAAs [83,84,85,86]. Self-expandable covered stents adapt to vessel tortuosity and seem to be more suitable than balloon-expandable stent for visceral arteries. For example, the Viabahn covered stent (Gore, Flagstaff, AZ, USA), which is PTFE coated, highly flexible and characterized by a peculiar lack of shape memory, was successfully used in 40 patients affected by VAAs (or visceral pseudoaneurysms) with permanent aneurysm exclusion and good stent patency rates at follow-up [87]. Balloon-expandable covered stents are usually more rigid and less suitable for the tortuous visceral arteries: in many cases, the artery is straightened to adapt to the stent losing its natural tortuous course. The use of a covered stent instead of embolization can be partially facilitated using extra-femoral approaches and coronary stent-grafts in particular situations. Transaxillary or transomeral approaches may overcome the issue of an unfavorable origin angle of the target artery from the aorta. The use of coronary stent-grafts, being more flexible and with smaller shafts than peripheral covered stents, can also extend the indication of covered stents to distally located VAAs [88], although the potentially treatable target artery is limited to a maximum diameter of 5–5.5 mm [48]. After covered stent placement, a double antiplatelet therapy is administered for 6 months, followed by lifelong aspirin. This protocol is important to achieve long-term patency of the stent. However, a slow and progressive covered stent occlusion due to intimal hyperplasia is not a problem, since aneurysm exclusion is maintained and the development of collateral vessels is able to provide organ revascularization and avoid ischemic complications [9]. The risk of covered stent infection, which is frequent in cases of mycotic aneurysm or pseudoaneurysm, should be prevented by antibiotic prophylaxis. A case of an infected visceral aneurysm successfully treated by an autologous-vein covered stent has been reported [89]. Cases of covered stent occlusion and subsequent extra-vascular migration have been recently reported in pseudoaneurysms, probably due to microenvironmental and post-surgical inflammatory conditions [90]. Covered stent placement may be more expensive than embolization, although it depends on the kind of embolization. However, in cases of fusiform or large saccular VAAs that have been treated with many detachable coils to obtain a high packing density, embolization can reach elevated costs. In complex aneurysms, a covered stent may be associated with coil embolization to prevent aneurysm revascularization.

## 5. Flow-Diverting Stent

Flow-diverting stents were initially developed for the treatment of cerebral aneurysms [91] and have been recently introduced in VAAs. Given a theoretical capacity of flow-diverters to obtain aneurysm thrombosis, preserving patency of its efferent branches [92,93], their use is preferable in districts at higher risk of ischemia, to maintain patency of both the target artery and side branches arising from the aneurysm. This strategy was initially developed in 2012 using a cobalt, multilayer, self-expandable stent (Cardiatis, Isnes, Belgium), specifically approved for peripheral and visceral aneurysms [94], but unsatisfactory findings were reported during the follow-up: only 60% of stent patency was observed at two years [95] and a case of disconnection of two overlapped stents [96] was reported. Flow-diverting stents for neuro-interventional use are characterized by more flexibility and lower-profile than peripheral covered stents. These features, in addition to higher porosity than bare metal stents, make them an ideal device for some VAAs with multiple side branches and with difficult anatomies (Figure 9). 

Aneurysms involving superior mesenteric artery or renal arteries with multiple side branches are particularly suitable to be treated with flow-diverting stents [97,98]. Their peculiar design promotes a progressive thrombosis of the aneurysm by reducing the flow at the aneurysm neck, determining a turbulence which leads toward an increased blood viscosity within the sac with final aneurysm exclusion [99]. A laminar flow into side branches including those arising from the aneurysm is preserved through the stent interstices, thanks to a neoendothelialization process [100]. The development of thrombosis, or aneurysm sac shrinkage due to depressurization, is longer compared to conventional techniques. In a recent meta-analysis, including 10 cohort studies with 220 patients with VAAs treated with flow-diverting stents, Zhang et al. reported side branches patency in 89% of the cases and aneurysm thrombosis/shrinkage in more than 90% of the cases [101]. Limits of flow-diverter stents are the elevated costs and the target artery diameter, which must not exceed 5–5.5 mm. As applied with covered stents, a double antiplatelet therapy for 6 months and lifelong aspirin should be recommended after flow-diverting stent placement.

## 6. Percutaneous (or Endoscopic) Approach

An alternative approach to the endovascular or surgical management of VAAs (or pseudoaneurysms) is represented by the percutaneous approach. A direct needle puncture can be performed under ultrasound or CT guidance when a saccular aneurysm with a narrow neck is located within solid organs without bowel interposition. The percutaneous approach may be performed alone or combined with an endovascular procedure, for example coil embolization [102]. Aneurysm thrombosis can be achieved carefully injecting thrombin, glue or EVOH liquid embolic agents under real-time imaging control [58,70,103]. Aneurysm sac opacification may be also obtained by direct contrast filling under fluoroscopic control, to check for the presence of efferent vessels. Endoscopic-ultrasonography (EUS) guided embolization may be rarely performed to treat aneurysms involving splenic, gastroduodenal, gastric or cystic arteries when visible at EUS [103,104]. The risk of a percutaneous approach without angiographic control is an extra-aneurysm accidental spread of thrombogenic material in non-target districts with ischemic complications. 

## 7. Visceral Artery-Related Considerations and Targeted Approaches

### 7.1. Celiac Trunk

Celiac trunk aneurysms are rare, accounting about 4% of VAAs. Endovascular options may be difficult especially if the aneurysm involves the origin of the artery due to the lack of a seal zone for a covered stent or the space to place coils proximally to the aneurysm. The presence of an adequate proximal and distal landing zone makes easier a covered stent placement or a coil embolization [105]. A fatal complication was recently described using a peripheral flow-diverting stent [106].

### 7.2. Splenic Artery

Splenic artery aneurysms represent 60% of all VAAs. A strong correlation with female gender, pregnancy and portal hypertension has been demonstrated [11]. Most of them (75%) are distally located, 20% in the middle third, 5% in the proximal tract. VAAs or pseudoaneurysms of the middle-third are often caused by pancreatitis [107]. In case of fusiform aneurysm, an endovascular repair with covered stent or a stent-assisted coil embolization may represent the first option but it is technically achievable only in case of proximal or intermediate location of the aneurysm, especially when dealing with markedly tortuous vessels. Coil embolization represents a safe and cost-effective alternative with low risk of ischemic complications, including in the case of splenic artery occlusion, thanks to short gastric collaterals [108]. Only embolization of hilar or intraparenchymal aneurysms with vessel sacrifice can cause splenic infarcts which are usually limited, not clinically significant and managed conservatively. Coil packing density is crucial to prevent aneurysm revascularization [109]. To enhance the embolic effect in large vessel such as splenic arteries, coils have been associated with adhesive or non-adhesive liquid embolic agents, like glue [110] and Squid [68].

### 7.3. Hepatic Artery

Hepatic artery aneurysms are the second most common entity, accounting for 20% of VAAs. The use of covered stent (flow-diverting stent, stent-assisted coil embolization), especially in case of proper hepatic artery involvement, is preferred to preserve hepatic artery flow and minimize the risk of severe organ ischemia. Coil embolization with sudden occlusion of the proper hepatic artery may cause hepatic infarction despite the liver dual vascularization and portal vein patency [111]. A fusiform aneurysm of common hepatic artery can be safely managed with coil embolization if collateral flow to the proper hepatic artery is guaranteed through the gastroduodenal artery [112]. Coil embolization of intrahepatic branches is generally performed without complications, thanks to a rich intraparenchymal anastomotic arterial network. 

### 7.4. Superior Mesenteric Artery

Superior mesenteric artery aneurysms account for about 5–7% of VAAs, usually involving the proximal part. An endovascular treatment may be burdened by the risk of ischemic complications which may require bowel surgical resection of the bowel segment involved following the procedure. Aneurysms located in the main trunk, which has multiple side branches, are usually treated with a short covered stent [113], a flow-diverting stent [97] or with stent-assisted coil embolization [114]. Aneurysms involving jejunoileal or colic proximal branches can be safely embolized with coils thanks to the presence of a rich anastomotic vascular network [115]. The use of particles is highly discouraged for the high risk of ischemic complications. 

### 7.5. Gastroduodenal and Pancreaticoduodenal Arteries

Gastroduodenal and pancreaticoduodenal artery aneurysms account for less than 5% of VAAs, usually due to pancreatitis but also atherosclerosis. Celiac trunk stenosis can cause hemodynamic alterations and pancreaticoduodenal artery aneurysms formation due to collateral overflow: an aneurysm coil embolization and celiac trunk angioplasty/stenting can be simultaneously performed [116]. In gastroduodenal or pancreaticoduodenal artery aneurysms, coil embolization is usually preferred to covered stenting due to the adequate collateral flow [117]. In selected cases, a percutaneous or endoscopic approach using thrombin or other embolic agents may be also performed [102,118]. 

### 7.6. Gastric Artery

Left gastric artery aneurysms account for about 4% of VAAs. Right gastric artery is rarely involved [119]. Endovascular management with coils or other embolic agents have been reported with low ischemic risk [120,121]. EUS treatment with glue has been also described [104].

### 7.7. Inferior Mesenteric Artery

Inferior mesenteric artery aneurysms are a rare entity accounting for less than 1% of VAAs. Coil embolization is usually the preferred endovascular option [122,123]. As for the superior mesenteric artery aneurysms, the use of particles is not recommended to avoid ischemic complications.

### 7.8. Renal Artery

Renal artery aneurysms, frequently associated to hypertension, account for 25% of VAAs. For narrow-necked saccular aneurysms coil embolization remains the most used endovascular technique to exclude the aneurysm and maintain regular flow in the parent artery [124]. For renal fusiform aneurysms involving distal branches, embolization with coils or liquid embolic agents (or particles) can be performed with acceptable minimal parenchyma sacrifice due to poor collateral formation [125]. In the case of wide-necked saccular aneurisms, fusiform aneurysms or complex aneurysms, covered stent or stent/balloon-assisted coil embolization represent important endovascular alternative options to simultaneously exclude the aneurysm and preserve the renal vascularization [21,50,126]. Neuro-interventional devices and flow-diverting stents have been recently employed and their use is growing [127,128].

### 7.9. Transplanted Arteries

Transplanted artery aneurysms or pseudoaneurysms after kidney, liver, pancreas transplantation may occur either in anastomosis site or in peripheral branches. Their endovascular management is similar to that of VAAs, including embolization, covered stent and stent-assisted coil embolization [129,130,131,132].

## 8. Conclusions

Endovascular management of VAAs includes different options. Selection of the best strategy depends on the involved visceral artery, aneurysm characteristics, the clinical scenario and the operator experience. Tortuosity of VAAs almost always makes embolization the only technically feasible option. Embolization with coils (alone or associated with other embolic agents) represents one of the most widely used endovascular techniques, especially in emergency settings, and has a moderate theoretical risk of ischemia. Endovascular repair with covered stent represents an ideal solution to exclude the aneurysm and preserve the parent artery, but it is feasible in less than 50% of cases. Stent- (or balloon-) assisted coil embolization is particularly suitable for complex aneurysms involving bifurcations or multiple ramifications. Flow-diverting stents and other neuro-interventional devices, have been recently introduced with promising results [56], but they are expensive and not widely available. Future perspectives include ever more accurate planning of the interventional procedures (CT reconstructions, artificial intelligence), enhanced and more controllable occlusive capacity of embolic agents (coils combined with EVOH non-adhesive embolic agents), more flexible and lower-profile stent-grafts without shape memory, suitable for the tortuous visceral arteries repair and less expensive neuro-interventional devices and flow-diverting stents. 

## Figures and Tables

**Figure 1 jcm-10-02520-f001:**
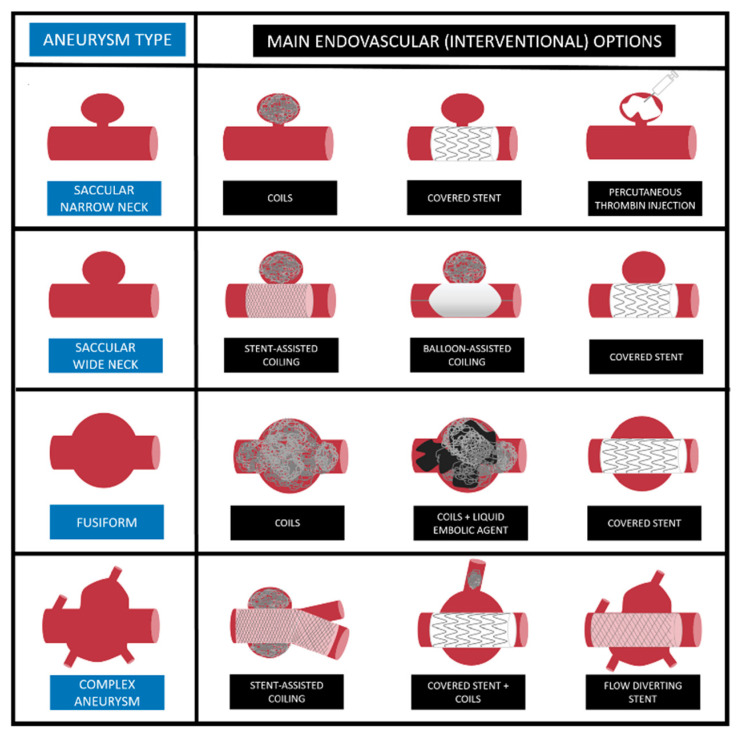
Schematic illustration of available endovascular techniques and main interventional options for treatment of VAAs based on aneurysm characteristics.

**Figure 2 jcm-10-02520-f002:**
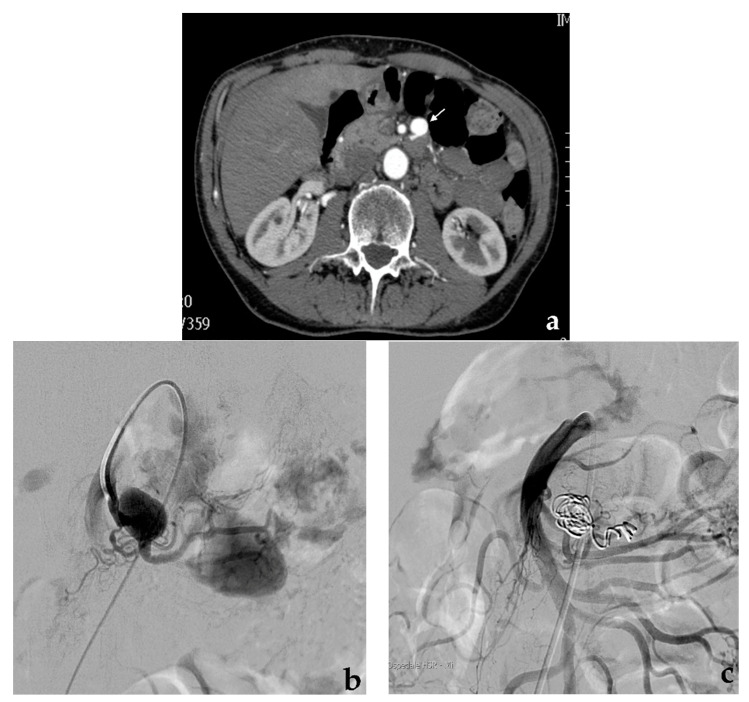
Coil embolization. (**a**) Preliminary contrast-enhanced CT (white arrow) and (**b**) diagnostic angiography show a fusiform aneurysm of a proximal jejunal branch of the superior mesenteric artery. (**c**) Final angiographic control shows the complete aneurysm exclusion after coil embolization performed occluding first the efferent vessel and then the aneurysm and the afferent vessel.

**Figure 3 jcm-10-02520-f003:**
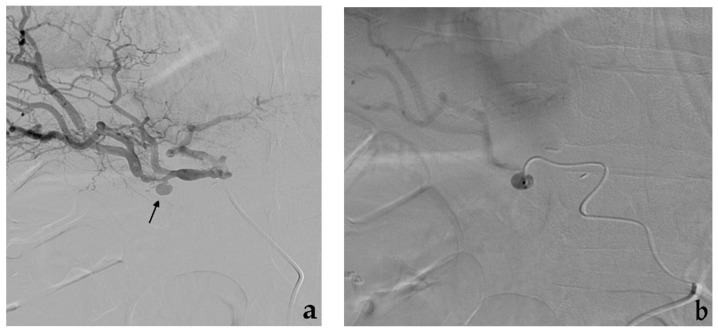
Coil embolization. (**a**) Diagnostic angiography shows a saccular aneurysm with a narrow neck of the right hepatic artery (black arrow). (**b**) Superselective aneurysm catheterization with a microcatheter and (**c**) progressive coils release. (**d**) Final angiographic control shows the complete aneurysm exclusion after coil embolization.

**Figure 4 jcm-10-02520-f004:**
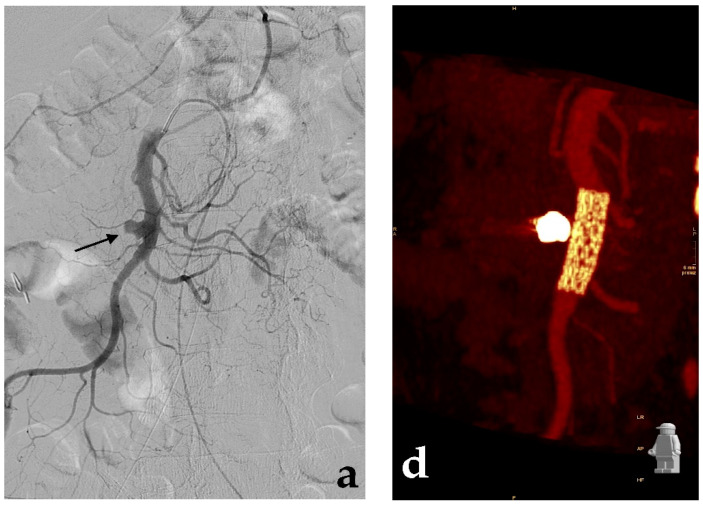
Stent-assisted coil embolization. (**a**) Diagnostic angiography shows a saccular aneurysm with a wide neck of the main trunk of the superior mesenteric artery (black arrow). (**b**) Using a dual femoral approach, aneurysm catheterization, uncovered stent release and aneurysm coil filling are subsequently performed (**c**) Final angiographic control and (**d**) CTA MIP reconstruction both show aneurysm exclusion with stent and superior mesenteric artery patency.

**Figure 5 jcm-10-02520-f005:**
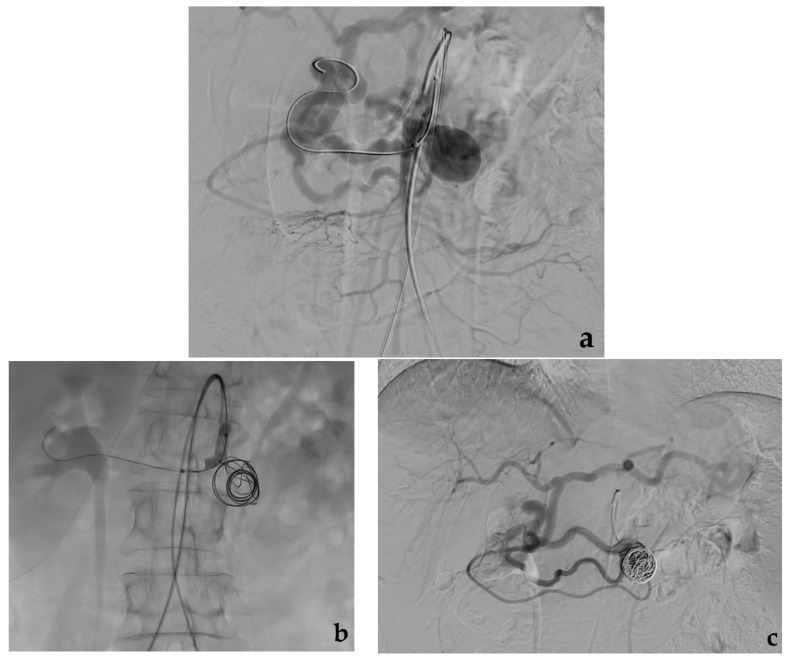
Balloon-assisted coil embolization. (**a**) Diagnostic angiography shows a saccular aneurysm with a wide neck of the pancreaticoduodenal artery. (**b**) Using a dual femoral approach, simultaneous balloon inflation and aneurysm coil filling. (**c**) Final angiographic control shows aneurysm exclusion without coil migration.

**Figure 6 jcm-10-02520-f006:**
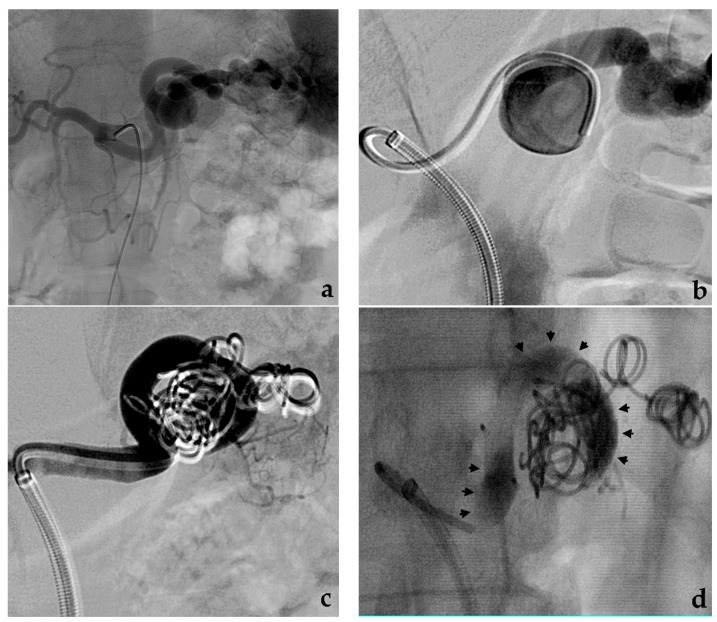
Embolization with coils and EVOH liquid embolic agent. (**a**) Diagnostic angiography shows a fusiform aneurysm of the splenic artery. (**b**) After selective catheterization of the aneurysm, (**c**) a progressive embolization of the efferent vessel and the aneurysm with coils and (**d**) the afferent vessel with Squid (black arrows) is performed. (**e**) Final angiographic control shows aneurysm exclusion with splenic artery occlusion.

**Figure 7 jcm-10-02520-f007:**
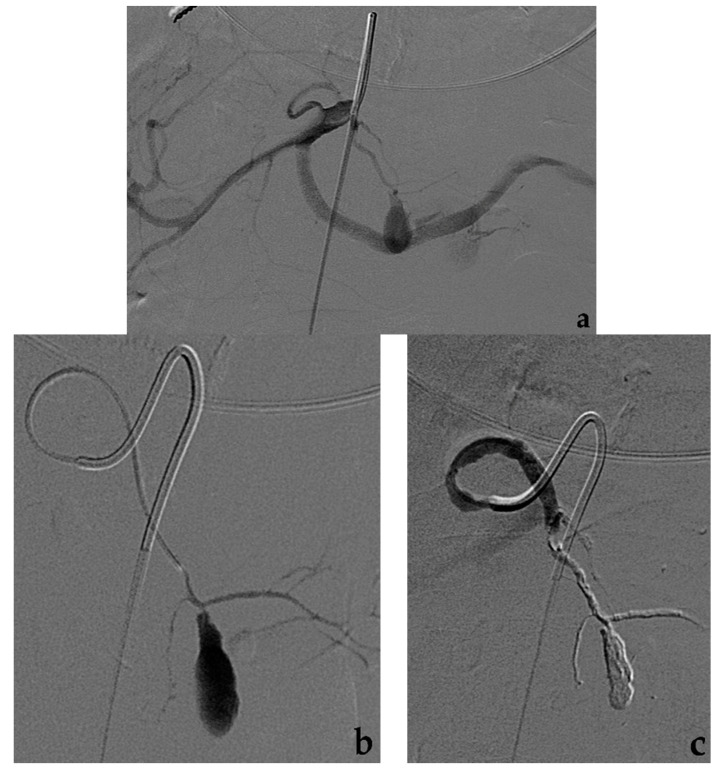
Embolization with EVOH liquid embolic agent alone. (**a**) Diagnostic angiography shows a pseudoaneurysm of a terminal branch of the left gastric artery. (**b**) Selective catheterization with a microcatheter of the afferent vessel to the pseudoaneurysm. (**c**) Final angiographic control after Onyx infusion shows complete pseudoaneurysm occlusion.

**Figure 8 jcm-10-02520-f008:**
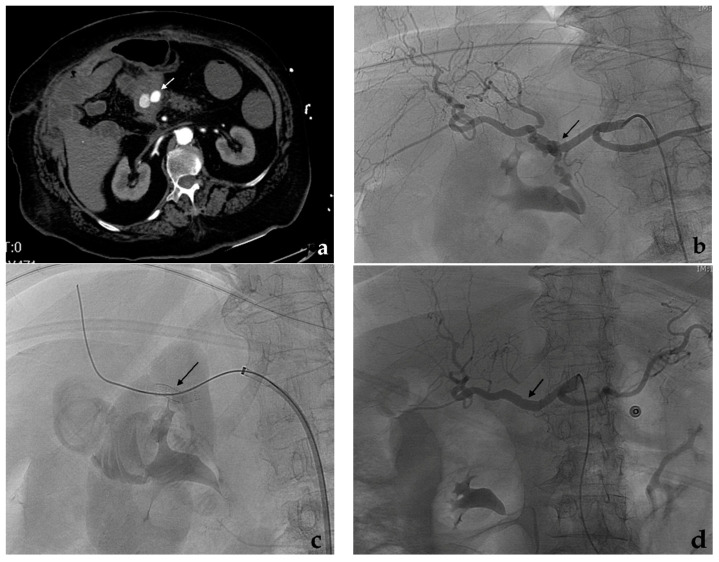
Covered stent. (**a**) Contrast-enhanced CT (white arrow) and (**b**) diagnostic angiography show a ruptured aneurysm of the common hepatic artery with contrast extravasation (black arrow). (**c**) After a self-expandable, covered stent (Viabahn, Gore) release (black arrow), (**d**) the final angiographic control shows the aneurysm exclusion with no more contrast extravasation and both stent and hepatic artery patency (black arrow).

**Figure 9 jcm-10-02520-f009:**
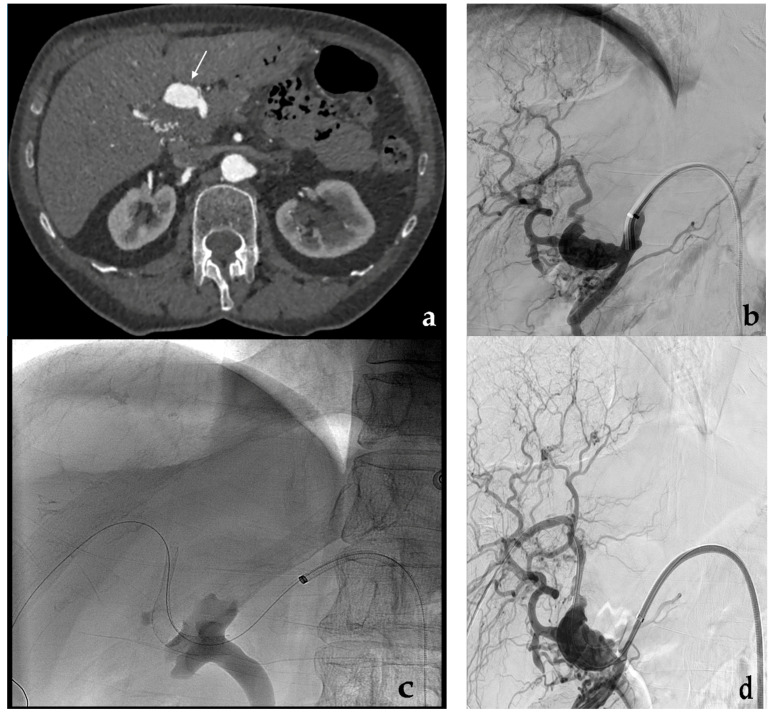
Flow-diverting stent. (**a**) Contrast-enhanced CT and (**b**) diagnostic angiography show an irregular aneurysm of the proper hepatic artery involving the origin of the gastroduodenal artery. (**c**) Flow-diverting stent (Surpass, Stryker) placement and (**d**) final angiographic control.

## Data Availability

No new data were created or analyzed in this study. Data sharing is not applicable to this article.

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
