# Peer review of "Visceral Artery Aneurysms Embolization and Other Interventional Options: State of the Art and New Perspectives"

_jcm, 2021, doi:10.3390/jcm10112520_

Round 1

Reviewer 1 Report

really nice draft

consider to improve the session of post-procedural imaging 

such as ceus discuss more extensively potential complications

Author Response

According to your comments, the session of post procedural imaging was improved also mentioning CEUS and potential complications were more extensively discussed.

The following sentences were added at the end of the paragraph 2: Metallic and bowel artifacts make also difficult a post embolization assessment with CDUS or contrast-enhanced ultrasonography (CEUS). CEUS can be useful for detect an ischemic complication of the target organ, such as the spleen, kidney or liver [38]” and at the end of the paragraph 3a: ”Clinically significant complications after VAAs embolization have been rarely reported. End-organ infarcts occurred in most cases have been conservatively traeted [50].” .

Moreover the following two references were added: 

  1. Pfister, K; Kasprzak, P.M.; Jung, E.M.; Müller-Wille, R.; Wohlgemuth, W.; Kopp, R.; Schierling, W. Contrast-enhanced ul-trasound to evaluate organ microvascularization after operative versus endovascular treatment of visceral artery aneu-rysms. Clin. Hemorheol. Microcirc. 2016, 64, 689-698.
  2. Etezadi, V.; Gandhi, R.T.; Benenati, J.F.; Rochon, P.; Gordon, M.; Benenati, M..; Alehashemi, S.; Katzen, B.T.; Geisbüsch, P. Endovascular treatment of visceral and renal artery aneurysms. J. Vasc. Interv. Radiol. 2011, 22, 1246-1253.

Reviewer 2 Report

Quite interesting and exaustive review of interventional therapy in risky abdominal visceral artery aneurysms. Advantages and strategies and clearly explained. The evidence of superiority of interventional techniques is very well defined, together with the close relationship between results and operator's experience. 

Author Response

I thank all the reviewers for their favorable comments and for truly appreciating our review article. 

Reviewer 3 Report

Venturini et al. present a well-written, comprehensive review of stent placement and embolization for arterial aneurysms in the visceral circulation. The authors present the circumstances that lead to the formation of such aneurysms and how they are usually asymptomatic and found incidentally.

The authors then present a beautifully illustrated series of paragraphs that describe the various ways stents are deployed and embolization is achieved via mechanical or chemical means. The use of balloons is also presented. The use of covered stents and flow diverting stents are also presented with excellent figures. Finally, percutaneous approaches are discussed.

The authors then discuss the advances and complications associated with the specific visceral vessels that can develop aneurysms. The paper is concluded with a reasonable, forward-looking summary.

Overall, the work is an easy read and provides an authoritative treatment of the subject. I have no important criticisms.

Author Response

(The authors gave the same response as above.)

Reviewer 4 Report

Authors performed a very interesting review on the treatment of visceral and renal aneurysms with IR procedures. The paper provide a nice overview of the topic, it is very easy to read, and very well organized. Pictures are wonderfus, with very nice cases, clearly illustrating each technique. References are adequate.

Author Response

(The authors gave the same response as above.)
